# Harnessing Wind Energy Potential in ASEAN: Modelling and Policy Implications

**Youngho Chang [1],\* and Han Phoumin [2]**

1    School of Business, Singapore University of Social Sciences, Singapore 599494, Singapore
2    Economic Research Institute for ASEAN and East Asia (ERIA), Jakarta 10270, Indonesia; han.phoumin@eria.org
\*    Correspondence: yhchang@suss.edu.sg

**Abstract:** This study examines whether and how harnessing more wind energy can decrease the cost of meeting the demand for electricity and amount of carbon emissions in the Association for Southeast Asian Nations (ASEAN) region, using the ASEAN integrated electricity trade model. Three scenarios are considered: a counterfactual business-as-usual (BAU) scenario, which assumes no wind energy is used; an actual BAU scenario that uses the wind-generation capacity in 2018; and a REmap scenario, which employs the wind-generation capacity from the Renewable Energy Outlook for ASEAN. Simulation results suggest that dispatching more wind energy decreases the cost of meeting the demand for electricity and amount of carbon emissions. However, these emissions increase during the late years of the study period, as the no- or low-emitting energy-generation technologies are crowded out.

**Keywords:** wind energy; power trade; counterfactual scenario; ASEAN

**JEL Classification:** Q41; Q42

## 1. Introduction

Wind energy can be considered the most promising renewable source for generating electricity. Currently, about 5.30 percent of the world's electricity is generated by wind power; 1429.6 terawatt-hours (TWh), of the 27,004.7 TWh of electricity generated in 2019, came from wind energy [1].

Table 1 shows the amount of the electricity generated in 2019. Coal has the largest share, followed by natural gas and hydroelectric power.

Among the electricity generated from renewable energy sources, wind energy has the largest share. Table 2 shows the amount of electricity generated by renewable energy in 2019. Wind covered slightly more than 50 percent of electricity generated by renewable energy.

For electricity generated from renewable sources, hydropower is the mode most utilised in the Association of Southeast Asian Nations (ASEAN) region followed by geothermal energy and solid biofuels. Wind energy comprised a very small share of the renewable energy in the region [2]. Similarly, hydropower had most of the installed capacity of renewable energy in the ASEAN region, and the capacity of wind generation was quite low [2].

ASEAN member countries have massive wind energy potential, however [2]. Across the region, there are many suitable sites where the speed of wind is ideal for harnessing electricity. Harnessing energy from wind can help provide clean energy at affordable prices and reduce carbon emissions. Yet, the potential utilisation rates are not great due to the intermittency of electricity generated from wind, a relatively high levelized cost of electricity (LCOE), and high balance-of-system costs. Financing renewable energy projects, including wind farms, is also a key barrier to improve the utilisation rate of the renewable potential [3].

**Table 1.** Electricity Generation by Fuels for the World. (Terawatt-hours).

|  | Oil | Natural Gas | Coal | Nuclear | Hydro | Renewables | Others | Total |
|---|---|---|---|---|---|---|---|---|
| Electricity | 825.3 | 6297.9 | 9824.1 | 2796.0 | 4222.2 | 2805.5 | 233.6 | 27,004.7 |
| Share (%) | 3.06 | 23.32 | 36.38 | 10.35 | 15.64 | 10.39 | 0.86 | 100.0 |

Note: 'Others' comprises sources not specified elsewhere. Source: [1].

**Table 2.** Renewable Electricity Generation in the World. (Terawatt-hours).

|  | Wind | Solar | Others | Total |
|---|---|---|---|---|
| Electricity | 1429.6 | 724.1 | 651.8 | 2805.5 |
| Share (%) | 50.96 | 25.81 | 23.23 | 100.0 |

Note: Others include geothermal, biomass, and other sources of renewable energy not already itemised. Source: [1].

Among ASEAN countries, Viet Nam has good sources of wind energy. Its current share of wind energy in its power generation mix in 2020 was 1.7 percent, lower than that of solar energy (12.8 percent). The potential of offshore wind energy there is 261 gigawatts (GW) (fixed) and 214 GW (floating). Fourteen offshore wind projects have been proposed, which total 28 GW [4]. Indeed, Viet Nam aims to install 12 GW to 15 GW of onshore wind energy and 10 GW to 12 GW of offshore wind energy by 2030 [5].

Some obstacles exist for Viet Nam's wind energy projects, however, especially offshore in terms of environmental, social, and technical constraints. The offshore sites include protected areas or essential habitats that house vulnerable marine species, birds, and bats. In addition, those sites include oil-related activities, energy and communications infrastructure, and aquaculture. They are commercial fishing grounds, tourism spots, and have great historical and cultural significance. To be fully utilised, they also must also clear technical constraints such as marine traffic, air traffic, and military use [4].

Using a cross-border power trade model in ASEAN [6], this study aims to demonstrate that renewable energy resources, especially wind energy, can help ensure energy sustainability and climate change adaptation. As a basis of evaluation for how wind energy can contribute to meet the electricity demand in ASEAN, it constructs a counterfactual business-as-usual (BAU) scenario in which no wind energy is used. Following this, an actual BAU scenario is used, using 2018 as the starting year. Finally, this study adopts a REmap scenario against which the counterfactual and actual BAU scenarios are evaluated to see how much wind energy can help meet the demand for electricity and reduce carbon emissions. An International Renewable Energy Agency (IRENA) study is used to show how renewable energy can contribute to the energy landscape in the ASEAN region, using 2025 as a target year [7].

The second section reviews prospects of harnessing wind energy and factors dragging this objective. The third section presents principles of harnessing wind energy that constitute the basis of the simulation model, and the fourth section discusses the methodology of this study, its key assumptions, and data. The fifth section discusses results of this study, and the sixth section presents policy implications derived from the study.

## 2. Harnessing Potential Wind Energy

### 2.1. Prospects

Huge potential exists for global wind power [8]. It can create more than 40 times the current worldwide consumption of electricity and more than 5 times the total global use of energy in all forms [9]. Wind energy can also bring non-energy benefits, as utilisation does not affect global temperature but does reduce carbon emissions and other air pollutants [10].

Some new technologies are currently exploring ways of harnessing energy from wind. A system that combines wind energy and hydropower in which the excess electricity generated from the wind farm is used to pump water from a lower tank to a higher level,

which was installed in the island of Ikaria in Greece, appears to be feasible for low-cost electricity production [11]. Navarre, a Spanish region, has exhibited how even small towns can become a big player in wind energy [12]. Some have also made efforts to harness energy from high-altitude wind where the speed of wind is faster and, hence, renders higher potential [13]. Moreover, power generated from offshore wind can be delivered via synoptic-scale interconnection, which appears to solve the underutilisation of wind power due to the fluctuation of electricity generated [14].

### 2.2. Drag Factors

Harnessing energy from renewable sources can have some negative environmental consequences. Indeed, the United Kingdom's Sustainable Development Commission was criticized for its failure to minimise the negative environmental consequences of wind energy such as noise, visual intrusion in sensitive landscapes, and bird strikes. The fair balancing of the advantages and disadvantages of harnessing wind energy in specific situations should be evaluated and after which wind energy should be utilised [15]. Wind farms also have a poor reputation; for example, it was reported that 40,000 birds in a year ran into wind turbine blades in the United States [16]. The modern type of wind turbine whose height is 125 m is almost as high as the London Eye, whose height is 135 m. After 16 years of litigation, relentless opposition from industrialists, and financial and political setbacks made a plan to build a wind farm in Cape Cod, Massachusetts, fail. The wind farm could have provided clean energy to 200,000 homes on Cape Cod and would have helped develop wind farms to nearby regions [17]. Financial viability also affects the development of wind energy, as, for example, the credit crunch drastically affected wind-energy projects in the United States during the Global Financial Crisis in 2008 [18].

In addition, the large-scale deployment of wind turbines appears to reduce wind speed and, in turn, lower turbine efficiency. The reduced wind speed eventually leads to set low generation limits [19].

Wind energy, especially onshore wind, is a mature technology that has achieved a certain level of reliability. However, the reliability, or load factor, is affected negatively by the age of the wind turbines. In the United Kingdom, the normalised load factor declined from about 24 percent during peak (i.e., age 1 year) to 15 percent at age 10 years, and 11 percent at age 15 years. The normalised load factor for Danish wind farms showed a similar decline—from 22 percent at age 1 year to 18 percent at age 15 years. Offshore Danish wind farms exhibited huge declines in their normalised load factors—from 39 percent at their peak to 15 percent at age 10 years [20].

### 2.3. Positive Signs of Harnessing Wind Energy

Wind turbines mounted on buildings appear to be feasible for reducing carbon emissions by contributing significantly to energy requirements in buildings. The aggregate electricity generated from these wind turbines range from 1.7 to 5.0 TWh per year and reduce carbon emissions by from 0.75 million to 2.5 million tons per year [21]. An energy company, Royal Dutch Shell, and an operator of oil tankers, Maersk, are also attempting to use wind power to cut tankers' fuel bills. They try to install the oil tanker with two 'rotor sails' to propel the vessel, which is one of other ideas of supplying energy to the oil tanker such as solar-powered sails and kites [22].

## 3. Principles of Wind Energy

### 3.1. Wind Energy as Kinetic Energy

Wind energy is kinetic energy that is transformed from potential energy. Scottish physicist William Rankine stated in 1881 that, "the object is gaining the potential to move 'by the occurrence of such changes, actual energy disappears and is replaced by Potential or Latent Energy'" [23]. Taking the definition of 'work' as the force multiplied by the distance moved in the direction of the force, the amount of energy harnessed from wind is determined by the speed of the wind and volume of air moved. When air-flow passes a

wind turbine at a given speed, a moving turbine constructs a hypothetical cylinder with the swept area as the length of the wind blade and the height as the speed of wind per second. The hypothetical cylinder captures air mass, which is kinetic energy, and is eventually transformed into electricity.

*3.2. Kinetic Energy in a Wind Turbine: Calculation*

Suppose a wind turbine with a diameter of 60 metres and a radius of 30 metres and the wind speed ($v$) of 9 metres per second.

- Swept area ($A$) is $\pi$ x $r^2 = \pi \times 30^2$
- Wind speed ($v$) is 9 metres per second (9 m/s)
- Volume of the cylinder ($V$) is $v \times A = 9 \times \pi \times 30^2 = 25{,}447$ cubic metres per second
- Density of air (the mass per cubic metre) is 1.29 kg per cubic metre
- Mass of air arriving per second ($m$) is $1.29 \times 25{,}447 = 32{,}827$ kg per second
- The kinetic energy of a mass $m$ moving with speed $v$ is $\frac{1}{2} mv^2 = \frac{1}{2} \times 32{,}827 \times 9^2 = 1{,}329{,}494$ joules per second = 1.33 megawatts (MW).

The principles of kinetic energy suggest that the longer the wind blade and the faster the wind speed, the more energy will be transformed from kinetic energy to electric energy (i.e., electricity). The modern type of wind turbine has a capacity of 1.8 MW [23].

*3.3. Economic Considerations of Wind Energy*

The cost of wind energy can be calculated as follows. This calculation is based on the information given in Boyle [23]:

The cost per unit ($g$) is expressed in Equation (1):

$$g = \frac{(C \times R)}{E} + M \tag{1}$$

where:

$g$ = the cost per unit of electricity generated;
$C$ = the capital cost of the wind farm;
$R$ = the capital recovery factor or the annual capital charge rate (expressed as a fraction);
$E$ = the wind farm annual energy output;
$M$ = the cost of operating and maintaining the wind farm annual output.

The required annual rate of return net of inflation ($R$) is expressed as:

$$R = [x/(1 - (1 + x))^{-n}] \tag{2}$$

where:

x = the required annual rate of return net of inflation;
n = the number of years over which the investment in the wind farm is to be recovered.

The annual energy output of the wind farm ($E$) is expressed as:

$$E = (hP_rF)T \tag{3}$$

where:

$h$ = the number of hours in a year (8760);
$P_r$ = the rated power of each wind turbine in kilowatts;
$F$ = the net annual capacity factor of the turbines at the site;
$T$ = the number of turbines.

The cost of operating and maintaining the wind farm annual output ($M$) is expressed as:

$$M = KC/E \tag{4}$$

where:

*M* = the operation and maintenance costs;
*K* = the factor representing the annual operating costs of a wind farm as a fraction of the total capital cost.

Generally, a wind turbine operates at only around 25 percent of turbine capacity due to inconsistent, imperfect wind. On better land-based wind sites, a capacity factor of 35 percent to 40 percent or more is achievable [23]. A wind turbine is quick to install, so it will be generating power before significant interest on capital. It is competitive with conventional power generation at sufficiently windy sites.

### 3.4. Unit or Levelised Costs of Wind Energy

A typical wind turbine has three parts: fiberglass blades, a standard gearbox, and a generator. Boyle [23] described the cost of a 600-kilowatt (kW) wind turbine in Denmark. Installation costs are $1800 to $2200 per kW, the turbine lasts about 20 years, the load factor is 25 percent, and the turbine generates 1,314,000 kilowatt-hours (kWh) per year. If a real discount or interest rate is assumed at 10 percent, the installation cost is $2000 per kW, or about $1,200,000. The unit or LCOE are $0.106 per kWh.

Table 3 presents the cost of generating electric power by various sources. The data are taken from estimated generation costs in the United States in 2017 for a comparison purpose.

**Table 3.** Estimated Cost of Generating Electric Power, 2017.

| Type | Cost (2010 $ per Megawatt-Hour) |
| --- | --- |
| Gas (all types) * | 66.1–127.9 |
| Hydro | 88.9 |
| Wind | 96.0 |
| Coal (all types) * | 97.7–138.8 |
| Geothermal | 98.2 |
| Advanced Nuclear | 111.4 |
| Biomass | 115.4 |
| Solar Photovoltaic | 152.7 |
| Solar Thermal | 242.0 |

* Includes carbon capture and sequestration. Source: [24].

Wind energy appears to be competitive with gas and coal. Moreover, the cost of electricity generated from wind is even lower than that of geothermal, although hydro is lower than wind. The cost competitiveness of wind in terms of power generation is also confirmed by the latest cost data provided by IRENA (Table 4).

**Table 4.** Weighted Average LCOE of Renewable Power Generation Technologies (kilowatt-hours).

| | Biomass | Geothermal | Hydro | Solar Photovoltaic | Concentrated Solar Power | Offshore Wind | Onshore Wind |
| --- | --- | --- | --- | --- | --- | --- | --- |
| 2010 | 0.076 | 0.049 | 0.037 | 0.378 | 0.346 | 0.161 | 0.086 |
| 2019 | 0.066 | 0.073 | 0.047 | 0.068 | 0.182 | 0.115 | 0.053 |

LCOE = levelized cost of energy. Notes: 1. The LCOE is the weighted average LCOE from utility-scale renewable power generation technologies from 2010 to 2019. 2. The fossil fuel LCOE range: = is $0.05–$0.18 per kilowatt-hour. Source: [25].

The LCOEs of geothermal and hydropower slightly increased in 2019 compared to 2010. The LCOEs of solar photovoltaic and concentrated solar power decreased immensely, while the LCOEs of offshore and onshore wind energy fell a small amount. Among various renewable power technologies, however, the LCOE of onshore wind energy is the second lowest after hydro. The LCOE of fossil fuels ranges from about $0.05 per kWh to about $0.18 per kWh [25]. Except for concentrated solar power and offshore wind energy, all

other renewable power-generation technologies have become competitive with fossil fuel power-generation technologies. The cost-competitiveness of wind energy is confirmed further if the cost of carbon disposal and the price of carbon are added to the LCOE.

Boyle [23] presented a comparison of the costs of various sources of electricity generation at a 10 percent discount rate. The cost included capital payments, operation and maintenance, fuel, carbon disposal, and carbon price. Fifteen power-generation technologies were considered: combined-cycle gas turbine, conventional coal, combined-cycle gas turbine with carbon capture and storage, coal with carbon capture and storage, nuclear-pressurised water reactor, roof-mounted solar photovoltaic thin-film panels, large biomass non-combined heat and power, run of river, reservoir hydro, onshore wind, offshore wind, tidal barrage, tidal stream, floating, and geothermal. The five lowest-cost technologies were run of river, reservoir hydro, combined-cycle gas turbine, onshore wind energy, and nuclear-pressurised water reactor. The LCOE of onshore wind is still higher than combined-cycle gas turbine. If a carbon price is added or the costs of carbon disposal for the combined-cycle gas turbine are included, then onshore wind energy is competitive with these technologies.

## 4. Methodology, Assumptions, and Data

This study explores how harnessing wind energy in the ASEAN region can reduce the cost of meeting the electricity demand and estimates the amount of carbon emissions that can be reduced.

### 4.1. Methodology

This study adopts the ASEAN integrated electricity grid model developed by Chang and Li [6] and modifies wind energy-related information. The objective of the integrated power trade model is to minimise the cost of meeting demand for electricity in the ASEAN region from 2018 to 2040. Costs have four components: capital cost, operation cost, transmission cost, and carbon cost. As it has an integrated electricity market and grid, power trade (i.e., the import of electricity) is allowed for up to 30 percent of domestic demand. Detailed description of the model is presented in Appendix A.

### 4.2. Assumptions

To meet domestic demand and trade surplus electricity, this study made some key assumptions. First, the total installed capacity of power generation in the region is greater or equal to the total demand for electricity in the region. Second, the total output of electricity generation in each country is constrained by the load factor of the installed capacity of all types of electricity generation in the county. Third, the electricity supply of all countries in the region to a certain country should be greater than or equal to the demand for electricity in that country. Fourth, the total supply of electricity from one country to all countries (including the country itself) in the region must be smaller or equal to the total available supply capacity of that country at a given time.

### 4.3. Data

This study updates the initial capacity given in Chang and Li [6] using the data taken from ASEAN Centre for Energy (ACE) [26] and International Renewable Energy Agency (IRENA) [25]. Figure 1 shows the initial installed capacity in ASEAN by plant type in 2018.

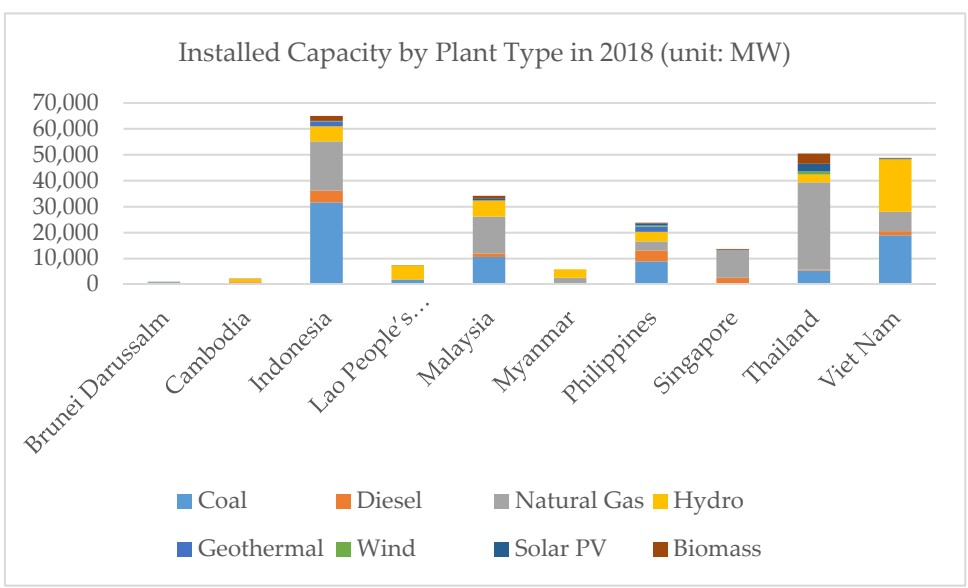

**Figure 1.** Installed Capacity by Plant Type in Association for Southeast Asian Nations (ASEAN), 2018. (Megawatts). PV = photovoltaic. Sources: [7] and [26].

*4.4. Scenarios*

This study establishes three scenarios: a counterfactual business-as-usual (BAU) scenario, an actual BAU scenario, and a REmap scenario. First, as the objective of this study is to estimate how much wind energy can help reduce the cost of meeting the electricity demand in the ASEAN region, it sets a counterfactual BAU scenario as a hypothetical base case. This assumes that no wind energy is used at all. In other words, there is no initial capacity of wind energy, and there is no added capacity of wind energy for the entire study period. This scenario presents the maximum possible contribution of wind energy to the cost of meeting the demand for electricity in the ASEAN region.

Second, an actual BAU scenario is set in 2018 in which the current initial capacity of wind energy is considered.

Third, a REmap scenario adopts the capacity of wind energy assumed in the REmap 2025 case in IRENA and ACE [7]. The REmap approach takes all available energy sources, including renewables, and considers energy supply and demand in power, heating, transport, and cooking. It aims to find a viable way of achieving the gap between the share of renewable energy under the reference case that is 17 percent and the target share of renewable energy for the region that is 23 percent. Full utilisation of potential wind energy is to be implemented in 2025 (Table 5).

**Table 5.** Expected Wind Capacity under REmap Scenario.

| Country | Wind Capacity (Megawatts) | Remarks |
|---|---|---|
| Brunei Darussalam | 0 | |
| Cambodia | 200 | |
| Indonesia | 2900 | |
| Lao People's Democratic Republic | 0 | |
| Malaysia | 100 | |
| Myanmar | 500 | |
| Philippines | 1100 | |

**Table 5.** *Cont.*

| Country | Wind Capacity (Megawatts) | Remarks |
|---------|---------------------------|---------|
| Singapore | 270 | Offshore wind |
| Thailand | 1800 | |
| Viet Nam | 5700 | |

Source: [7].

As stated previously, Viet Nam is expected to utilise its huge potential of wind energy and install the largest capacity of wind energy (5700 MW) among the 10 ASEAN countries. Indonesia is next at 2900 MW, and Thailand and the Philippines are in third and fourth with installed wind capacity of 1800 MW and 1100 MW, respectively.

## 5. Results, Discussions, and Policy Implications

### 5.1. No Wind Energy

The counterfactual BAU scenario presents the highest cost of meeting electricity demand in the ASEAN region and has the largest carbon emissions.

5.1.1. Cost of Electricity Generation in ASEAN Countries

When all capacities of wind energy are intentionally removed from the available technologies, three distinct trends emerge compared to the actual BAU case (Table 6). First, more low-cost technologies, such as hydropower, are used across many countries from 2026 to 2040. Second, renewable energy technologies, such as geothermal energy for Indonesia and the Philippines, are dispatched. Along with early utilisation of geothermal energy, more biofuel energy is utilised in Singapore. The Philippines appears to tap into biofuel energy as well. Third, more carbon-intensive and costly carbon-generation technologies, such as coal with carbon capture and storage and gas with carbon capture and storage, appear to be dispatched later in 2036 and 2040.

**Table 6.** Cost of Meeting Electricity Demand in the ASEAN Region ($ billion).

| Scenarios | Cost | Difference |
|-----------|------|------------|
| Counterfactual BAU | 421.05 | - |
| BAU | 418.20 | 0.7% |
| REmap | 409.36 | 2.8% |

BAU = business as usual. Source: Authors.

When ASEAN countries utilise wind energy, however, the cost of meeting electricity demand in the region is lowered by about 0.7 percent. The share of wind energy, out of the total installed generation capacity in the ASEAN region, is about 0.8 percent. The cost of wind energy is almost the same as the share of installed generation capacity. Figure 2 presents the cost of meeting electricity demand in ASEAN countries.

The total cost of meeting the demand for electricity in the ASEAN region is $421.05 billion if no wind energy is utilised at all, i.e., the counterfactual BAU scenario. Under the BAU scenario in which the current level of wind energy is assumed, the total cost is $418.20 billion, about 0.7 percent lower than that of the counterfactual BAU scenario. The total cost of the counterfactual BAU scenario is $421.05 billion while that of the REmap scenario is $409.36 billion. The difference between the counterfactual scenario and the REmap scenario is 2.8 percent, which is more than three times the difference between the cost of the counterfactual scenario and BAU scenario, if the capacity of wind energy assumed under the REmap scenario of IRENA and ACE [7] is to be fully utilised from 2025.

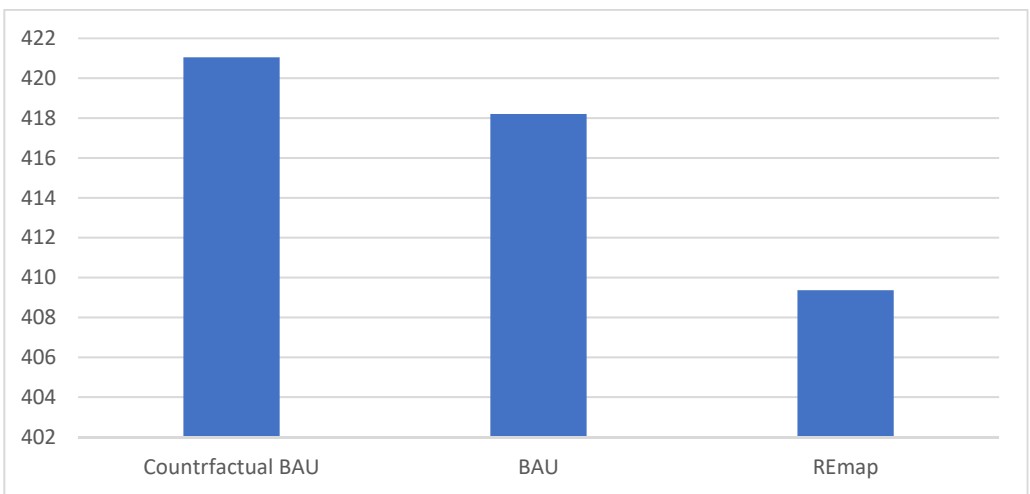

**Figure 2.** Total Cost of Meeting the Demand for Electricity in ASEAN. ($ billion). BAU = business as usual. Source: Authors.

5.1.2. Carbon Emissions

The difference in carbon emissions between the counterfactual scenario and REmap scenario is interesting (Figure 3). The difference in the quantity ranges from 0.62 million tons in 2039 to 29.71 million tons in 2025, mostly because new capacity of wind energy is assumed to be installed in 2025. Excluding this, the next highest difference is achieved in 2028. The amount of carbon emissions under the counterfactual BAU scenario is slightly higher than the REmap scenario in 2038, probably due to the lower capacity of hydro, which is added in 2038.

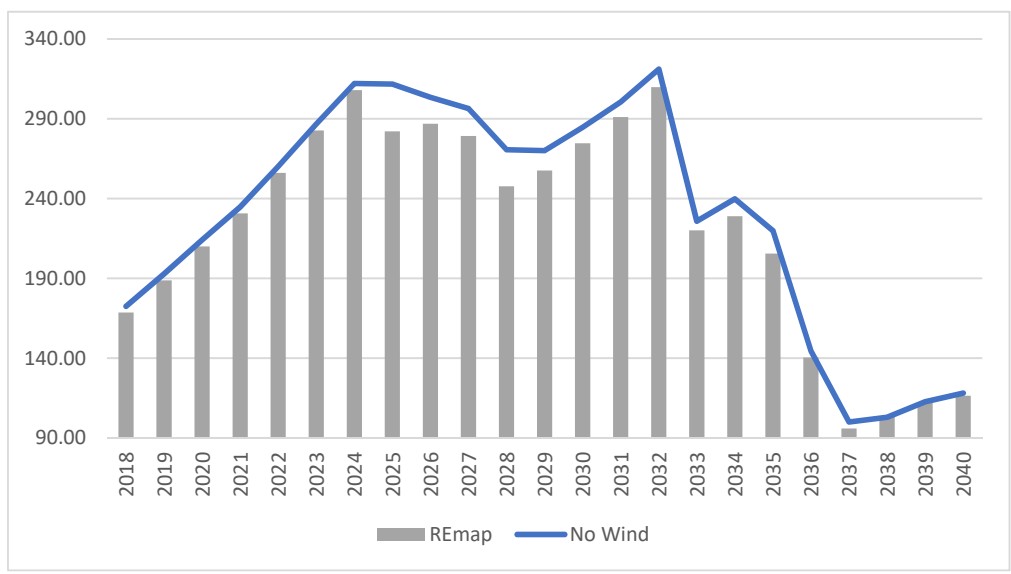

**Figure 3.** Trajectory of Carbon Emissions under Counterfactual BAU and Remap (million tons). BAU = business as usual. Source: Authors.

Thus, utilising more wind energy could reduce carbon emissions further. The simulation of the REmap scenario shows that a few countries in ASEAN, such as Brunei Darussalam, Malaysia, Singapore, and Thailand, appear to fully utilise their potential for wind energy. If other countries are able to harness their potential for wind energy, then the reduction in carbon emissions could be even larger.

### 5.2. Actual Business-as-Usual Scenario and REmap Scenario

A more realistic evaluation of how wind energy can reduce carbon emissions is shown by comparing the simulation results of the actual BAU scenario with those of the REmap scenario in which the full utilisation of potential for wind energy is expected to start from 2025. Figure 4 presents possible amount of carbon emissions reduced in the REmap scenario.

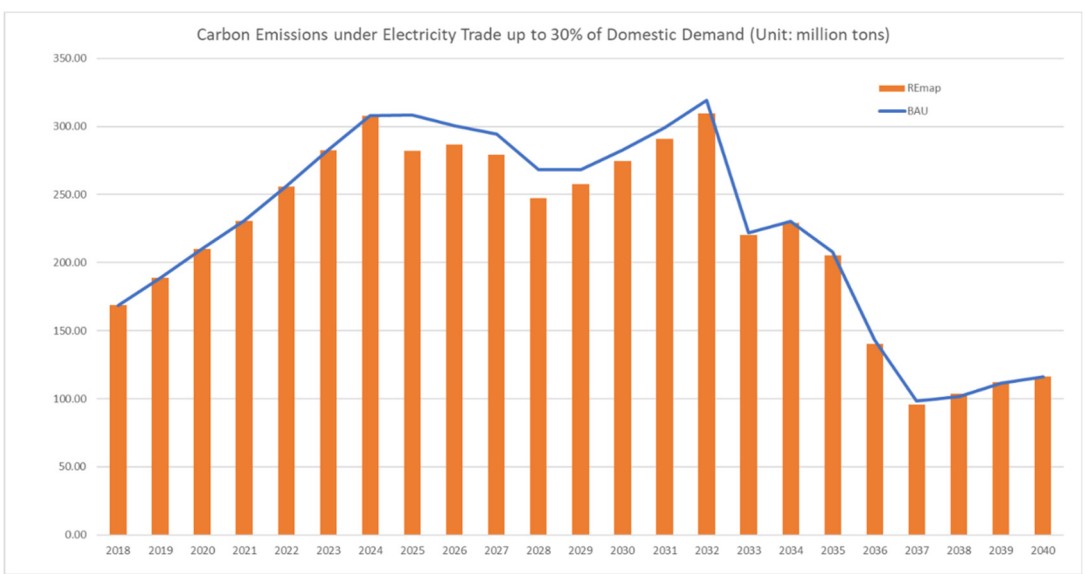

**Figure 4.** Reductions in Carbon Emissions under REmap Scenario (million tons). BAU = business as usual. Source: Authors.

The difference in the quantity of carbon emissions ranges from 1.44 million tons in 2034 to 26.22 million tons in 2025, mostly because new capacity of wind energy is assumed to be installed in 2025. Excluding this, the next highest difference is achieved in 2028. Carbon emissions under the REmap scenario appear to be higher than those under the actual BAU scenario during the last three years of the study period, caused by less hydro capacity during those years.

### 6. Conclusions and Policy Implications

ASEAN countries have good potential to harness wind energy, especially Viet Nam. Wind energy, however, is not commensurate with the degree of potential capacity. The intermittency of wind and high system costs are the main reasons for low development.

This study found that there would be 0.7 percent higher costs in meeting the demand for electricity in ASEAN countries if no wind energy was utilised. The costs of meeting the demand for electricity in ASEAN under the REmap scenario appear to be about 2.8 percent lower than that of the counterfactual scenario. As expected, the amount of carbon emissions from both the actual BAU scenario and the REmap scenario are lower than that of the counterfactual scenario, especially from 2025 when wind energy is extensively harnessed.

The trajectories of carbon emissions exhibit a visible gap between the counterfactual BAU scenario and REmap scenario from 2025 to 2032 and a lesser visible difference toward 2040. All three scenarios show that the level of carbon emissions would peak around the early 2030s when carbon-emitting power-generation technologies are more extensively dispatched to meet the increasing demand for electricity in the ASEAN region.

The REmap scenario shows that both the cost of meeting the demand for electricity and amount of carbon emissions decrease compared to the counterfactual BAU scenario and actual BAU scenario. However, the amount of carbon emissions appears to increase during later periods, as low- or no-carbon-emitting technology is crowded out. Considering the possible reverse in the trajectories of carbon emissions, whether the added capacity of wind energy will increase the amount of carbon emissions needs to be evaluated. If the

reversal in the amount of carbon emissions appears to be the case, then such a case should not proceed.

This study draws a few policy implications from the findings presented above.

First, as shown in the REmap scenario, more wind capacity appears to accelerate the decreasing trend of carbon emissions. Wind energy should, thus, be promoted in ASEAN countries. As the cost of harnessing wind energy is expected to decrease further, more wind energy will lower the cost of meeting the electricity demand in ASEAN.

Second, the amount of carbon emissions could be larger when more wind capacity is dispatched, although the cost of meeting the demand for electricity will decrease. When a decision to add more wind capacity is made, a rigorous evaluation should proceed to determine whether the wind capacity will crowd out no- or low-carbon-emitting technologies, such as hydro, and eventually increase carbon emissions in the long term.

Third, harnessing more viable renewable energy power-generation technologies in the ASEAN region could decrease the level of carbon emissions. It is uncertain, however, if dispatching more of such technologies would decrease the costs of meeting the demand for electricity. ASEAN countries need to decrease the costs of renewable energy power-generation technologies, therefore, through more research and development.

Harnessing renewable energy power-generation technologies is not immune from damaging the environment and can have negative repercussions on the economy, as identified in Viet Nam's development of offshore wind energy. Thus, ASEAN must evaluate possible negative impacts of harnessing renewable energy on the environment and economy.

This study can be improved with more detailed data such as country-specific peak and offpeak demand. Such data can make the simulation study produce more realistic results. The results, in turn, will present more effective and relevant policy implications.

**Author Contributions:** Conceptualization, Y.C. and H.P.; methodology, Y.C. and H.P.; software, Y.C.; validation, Y.C. and H.P.; formal analysis, Y.C.; investigation, Y.C.; resources, Y.C.; data curation, Y.C.; writing—original draft preparation, Y.C.; writing—review and editing, Y.C. and H.P.; visualization, Y.C.; supervision, Y.C. and H.P.; project administration, Y.C. and H.P.; funding acquisition, H.P. All authors have read and agreed to the published version of the manuscript.

**Funding:** This research was partially funded by Economic Research Institute for ASEAN and East Asia (ERIA).

**Institutional Review Board Statement:** Not applicable.

**Informed Consent Statement:** Not applicable.

**Data Availability Statement:** Not applicable.

**Conflicts of Interest:** The authors declare no conflict of interest.

## Appendix A

*Appendix A.1. Model Description*

This study adopts a dynamic linear programming framework in power generation first developed by Turvey and Anderson [27] and later adapted by Chang and Tay [28] and Chang and Li [6]. In the latest study by Chang and Li [6], significant extensions of the original models were made. A new country dimension was added to allow an international framework with cross-border electricity trade. The new model also added the cost of cross-border power transmission as well as transmission loss into account. Carbon emissions from power generation as well as the carbon cost of power generation were explicitly considered. The model was solved using General Algebraic Modelling System (GAMS).

This section documents how the capital expenditure (CAPEX) and operational expenditure (OPEX) of a certain type of power generation is represented in the model and how carbon emissions and cross-border transmission cost are explicitly represented in the

model. Following this, it presents the objective function and various constraints to make the power trade model work.

*Appendix A.2. CAPEX*

The capital expenditure (CAPEX) of a certain type of power generation capacity at a certain point of time is modelled as follows. The total capital cost of a certain type of power generation capacity during the period of this study is expressed as $\sum_{i=1}^{I} \sum_{v=1}^{T} \sum_{m=1}^{M} c_{miv} * x_{miv}$. $x_{miv}$ is the capacity of plant type m, vintage v, in country i. Vintage indicates the time a certain type of capacity is built and put into use. $c_{miv}$ is the corresponding capital cost per unit of capacity of the power plant. For simulation purpose and consistency in presentation with the other cost terms, a time dimension to the equation besides the vintage dimension is added. This allows that the capital cost is amortized using a capital recovery factor.

*Appendix A.3. OPEX*

The operational expenditure (OPEX) of a certain type of power generation capacity at a certain point of time is modelled as follows. The operation cost of a certain type of power generation capacity in year t is expressed as $\mathrm{Opex}(t) = \sum_{i=1}^{I} \sum_{j}^{J} \sum_{v=-V}^{t} \sum_{p=1}^{P} \sum_{m=1}^{M} F_{mitv} * u_{mijtvp} * \theta_{jp}$. $u_{mijtvp}$ is the power output of plant m, vintage v, in year t, country i, block p on the load, and exported to country j. $F_{mitv}$ is the corresponding operating cost that varies with v, and $\theta_{jp}$ is the time interval of load block p within each year in the destination country.

*Appendix A.4. Carbon Emissions*

Carbon emissions of different types/technologies of power generation capacity are modelled as follows and the cost of carbon emissions is explicitly considered. The amount of carbon emissions produced is expressed as $\sum_{m=1}^{M} \sum_{i=1}^{I} \sum_{j=1}^{J} \sum_{v=-V}^{T} u_{mijtvp} * \theta_{jp} * ce_m$, and the carbon cost in year t is $\mathrm{CC}(t) = cp_t * \left( \sum_{m=1}^{M} \sum_{i=1}^{I} \sum_{j=1}^{J} \sum_{v=-V}^{T} u_{mijtvp} * \theta_{jp} * ce_m \right)$. $ce_m$ is the carbon emissions per unit of power plant capacity of type j plant, and $cp_t$ is the carbon price per unit of carbon emissions in year t.

*Appendix A.5. Cross-Border Transmission Cost*

The costs of cross-border transmission come in two forms—a tariff and transmission loss. The tariff is paid to recover the capital investment and operational cost of the grid line. The transmission loss could be significant if the distance of transmission is long and is explicitly considered in Equation (4). The tariff of transmission, $tp_{i,j}$, is the unit MWh transmission cost of power output from country i to country j. The total cost of cross-border power transmission in year t, using $tp_{i,j}$, is expressed as $\mathrm{TC}(t) = \sum_{i=1}^{I} \sum_{j=1}^{J} \sum_{v=-V}^{T} \sum_{p=1}^{P} u_{mijtvp} * \theta_{jp} * tp_{i,j}$.

*Appendix A.6. Objective Function*

The objective of the power trade model is to minimize the total cost of electricity during the period of this study. The objective function is written as:

$$\mathrm{obj} = \sum_{i=1}^{I} \sum_{v=1}^{T} \sum_{m=1}^{M} c_{miv} * x_{miv} + \sum_{t=1}^{T} \{\mathrm{Opex}(t) + \mathrm{CC}(t) + \mathrm{TC}(t)\} \qquad (A1)$$

*Appendix A.7. Constraint Conditions*

There are several constraints that are required to optimize the above objective function.

Equation (A2) shows a first set of constraints, which requires total power capacity to meet total power demand in the region. $Q_{itp}$ is the power demand of country $i$ in year $t$ for load block $p$.

$$\sum_{i=1}^{I}\sum_{j=1}^{J}\sum_{m=1}^{M}\sum_{v=-V}^{t} u_{mijtvp} \geq \sum_{i=1}^{I} Q_{itp} \tag{A2}$$

The second one, shown in Equation (A3), states the constraint of load factor $lf_{mi}$ of each installed capacity of power generation. $kit_{mi}$ is the initial vintage capacity of type $m$ power plant in country $i$.

$$u_{mijtvp} \leq lf_{mi} * (kit_{mi} + x_{miv}) \tag{A3}$$

The third constraint, shown in Equation (A4), says that power supply of all countries to a certain country must be greater than the country's power demand. $tl_{i,j}$ is the ratio of transmission loss in cross-border electricity trade between country $i$ and country $j$.

$$\sum_{j=1}^{J}\sum_{m=1}^{M}\sum_{v=-V}^{t} u_{mijtvp} \cdot tl_{ij} \geq Q_{itp} \tag{A4}$$

The fourth constraint, shown in Equation (A5), states that total supply of power of one country to all countries (including itself) must be smaller than the summation of the country's available power capacity at the time.

$$\sum_{j=1}^{J} u_{mijtvp} \leq \sum_{m=1}^{M}\sum_{v=-V}^{t} lf_{mi} * (kit_{mi} + x_{miv}) \tag{A5}$$

The fifth constraint, shown in Equation (A6), is capacity reserve constraint. $pr$ is the rate of reserve capacity as required by regulation. Moreover, $p = 1$ represents the peak load block.

$$\sum_{i}^{I}\sum_{m=1}^{M}\sum_{v=-V}^{t} lf_{mi} * (kit_{mi} + x_{miv}) \geq (1 + pr) * \sum_{i}^{I} Q_{it,p=1} \tag{A6}$$

Hydro-facilities have the so-called an energy factor constraint as shown in Equation (A7). $ef_{mi}$ is the energy factor of plant type $m$ in country $i$. Other facilities have $ef = 1$.

$$\sum_{p=1}^{P}\sum_{j=1}^{J} u_{mijtvp} \leq ef_{mi} * (kit_{mi} + x_{miv}) \tag{A7}$$

Lastly, development of power generation capacity faces resource availability constraint, which is shown in Equation (A8). $XMAX_{mi}$ is the type of resource constraint of plant type $m$ in country $i$.

$$\sum_{v=1}^{T} x_{miv} \leq XMAX_{mi} \tag{A8}$$

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
