# Peer review of "Harnessing Wind Energy Potential in ASEAN: Modelling and Policy Implications"

_sustainability, doi:10.3390/su13084279_

Round 1
Reviewer 1 Report
The authors wrote on the policy implications of harnessing more wind energy in the ASEAN electricity supply system. The authors introduced the work and cited relevant literature in the report. They also discussed their results.
The authors need to do a thorough revision of the work to make it more valuable to the reader. Apart from the fact that the manuscript does not seem to contain much new information, the authors did not include the model for the research in the report. Also, some data adopted in the research seem not to be reliable and needs to be reviewed. More of my comments are on the highlighted portions of the manuscript (attached).
I will also suggest that the authors employ the services of a professional English Language translator to assist in revising the work to make the report better and on point. I offer this suggestion because the language style adopted for this report is weak and the report is verbose, making it unimpressive to read.

Reviewer 2 Report
The authors get a good idea to develop their research. However, they should improve each of the sections of this paper (See revised document attached). I think some basic citations and references are missing, that could be added in the introductory section and to expand the discussion of the results.
Best regards and health!
Author Response
We do not find the revised document that the reviewer said 'attached'. But we attached our revised manuscript.

Reviewer 3 Report
The paper undertakes an important and current research problem, however, the title promises much more than the content. That is why it seems to be a bit disappointing.
The paper is not well-grounded in theory. There is no theoretical basis. A literature review could be more focused on presenting the results of research by other researchers. The research gap in the studies should be clearly defined. The issues presented in the current version of the article could be a starting point for further empirical analysis. I would consider introducing the literature review as one section of the paper.
The introductory section has to catch the attention and interest of the readers. The authors should improve this part and present it more clearly and attractively. I would consider developing the importance of the subject, aim and contribution of the paper.
There are no research hypotheses and research questions in the study. They should be introduced.
The research methods and frameworks should be appropriately designed and presented in the paper. In such studies, the reviewed literature should be significantly enriched and based mainly on the latest research results.
The discussion on the results should be developed and more extensively linked to the literature. Moreover, the considerations should be complemented by the limitations of the research and the directions for further research.
The content should be logically ordered. I suggest developing scientific arguments.
Minor comment: I would carefully check the references used in the paper with the list compiled at the end of the study.
Author Response
Following the comment, we re-wrote our paper.
We revised our paper and attached here. Thank you!

Round 2
Reviewer 1 Report
The authors made efforts in addressing some of the grammatical concerns raised, but my major concerns are not yet addressed.
For instance, The data in Table 3 is captioned "U.S. New Generation in 2017" but was taken from a 2015 publication. This data is either projected or estimated data and is not suitable to be used in the context that it serves in this manuscript.
Also, I pointed out that the description of the methodology is inadequate. It is not enough to refer readers to another for the description of the methodology adopted in a report. Here, the authors mentioned that they did optimization. I need to see the objective function and the constraints of the optimization model so that I can decide on the model's validity and suitability in this study.
Furthermore, it essential to ensure that any conclusion and policy implication derived from the study must have a direct correlation with the results of the study.
Author Response
Thank you very much for insightful and valuable comments.
The data in Table 3 is estimated cost of U.S. new generation in 2017 that is presented as a comparison purpose.
Detailed description of the model is presented in Appendix.
Conclusion and policy implications are derived from the results of this study.
Please refer to the revised version attached.

Reviewer 2 Report
The authors did not find my file with suggestions for improvement. I see that the authors have improved their paper according to suggestions for improvement from other reviewers. I resend my file so that the authors can make the appropriate changes.
Best regards.

Author Response
Thank you very much for your productive comments.
The revised version, which is edited by a professional English editor, has reflected those comments.
We hope the revised version duly reflected the comments given based on the earlier version.
Please refer to the revised version attached.

Reviewer 3 Report
Main comments: In my opinion, the purpose of the article still needs to be emphasized. Moreover, I would also introduce research questions. In my opinion, the conclusions still need to be developed and limitations of the research, as well as directions of further research should be added.
Minor comments: I am not an expert on the English language, but I recommend the Authors to check language and style.
Author Response
Thank you very much for your productive comments.
The revision was done to reflect all the comments including research questions, conclusion and limitations.
The revised version has been edited by a professional English editor.
Please refer to the revised version attached.

Round 3
Reviewer 1 Report
With reference to my previous comments, the authors have addressed the concerns I raised and the manuscript retains all the positive attributes I identified earlier.